# Effects of insulin resistance and β-cell function on diabetic complications in Korean diabetic patients

**Do Kyeong Song, Young Sun Hong, Yeon-Ah Sung, Hyejin Lee***

Department of Internal Medicine, Ewha Womans University School of Medicine, Seoul, Korea

* hyejinlee@ewha.ac.kr

**Data Availability Statement:** All relevant data are within the manuscript and its Supporting Information files.

## Abstract

### Background

Diabetes mellitus is characterized by insulin resistance (IR) and dysfunctional insulin secretion from pancreatic β-cells. However, little research has been conducted on the relationship between IR and β-cell function in relation to diabetic complications among Korean diabetic patients. This study aimed to examine the differential associations between IR and β-cell function and various diabetic complications among Korean diabetic patients.

### Methods

The analysis employed a common data model (CDM). IR and β-cell function were quantified using the homeostasis model assessment for insulin resistance (HOMA-IR) and β-cell function (HOMA-β), respectively. Hazard ratios for diabetic nephropathy, diabetic retinopathy, and cardiovascular disease (CVD) events were calculated.

### Results

The study cohort consisted of 2,034 diabetic patients aged over 20 years who visited EUMC between January 2001 and December 2019. Among diabetic patients in the highest quartile of HOMA-IR, the adjusted hazard ratio for total CVD events was 1.76 (95% confidence interval [CI], 1.20–2.57) compared with those in the lowest quartile of HOMA-IR ($P = 0.004$). In contrast, diabetic patients in the lowest quartile of HOMA-β exhibited an adjusted hazard ratio of 3.91 (95% CI, 1.80–8.49) for diabetic retinopathy compared to those in the highest quartile of HOMA-β ($P = 0.001$).

### Conclusion

Insulin resistance and β-cell function exhibited different associations with diabetic complications among Korean diabetic patients. Specifically, lower β-cell function was associated with an increased risk of diabetic retinopathy, whereas higher IR was associated with an increased risk of CVD events. Individuals with pronounced IR should prioritize CVD prevention measures, and those with significant β-cell dysfunction may benefit from early, intensive surveillance for diabetic retinopathy.

**Funding:** This work was supported by the Ewha Womans University Research Grant of 2021. The funder had no role in study design, data collection and analysis, decision to publish, or preparation of the manuscript.

**Competing interests:** We declare that we have no conflict of interests.

## Introduction

Diabetes mellitus is a multifactorial condition characterized by insulin resistance (IR) and insufficient insulin secretion from pancreatic β-cells in response to glucose. Patients with diabetes exhibit heterogeneous phenotypes and diverse risks for complications, drug responses, and disease progression [1]. Approximately 75–85% of diabetic patients meet the traditional criteria for type 2 diabetes, but many show characteristics of both insulin resistance and β-cell dysfunction [2, 3]. Despite risk factor management, the incidence of diabetic complications remains high and understanding of the variation of diabetic complications is lacking [4].

To better identify patients at elevated risk for complications upon diagnosis and to facilitate personalized treatment approaches, a refined classification system was proposed in Sweden. This system delineated five novel subgroups of newly diagnosed patients: severe insulin-deficient diabetes (SIDD), severe insulin-resistant diabetes (SIRD), severe autoimmune diabetes (SAID), mild obesity-related diabetes (MOD), and mild age-related diabetes (MARD). Risks for specific complications also varied; SIRD was more likely to develop diabetic kidney disease, while SIDD was more prone to diabetic retinopathy [5]. Additionally, IR served as an independent predictor for incident cardiovascular diseases (CVD) among subjects with type 2 diabetes in the Verona Diabetes Complications Study [6]. Diabetic retinopathy was also found to be linked to reduced β-cell responsiveness in Caucasian individuals with type 2 diabetes [7].

The clinical characteristics of diabetes vary depending on ethnicity, dietary habits, and lifestyle factors. Historically, type 2 diabetes in East Asians has been primarily characterized by β-cell dysfunction, reduced adiposity, and a younger age of onset compared to Caucasians with the same condition [8, 9]. High levels of homeostasis model assessment of insulin resistance (HOMA-IR) have been independently associated to a higher prevalence of cardiometabolic disorders compared to low homeostasis model assessment of β-cell function (HOMA-β) in Chinese adults [10]. In a Japanese retrospective cohort study, patients with SAID or SIDD were found to be at greater risk for diabetic retinopathy, while those with SIRD had an elevated risk for diabetic kidney disease [11]. Genetic and environmental factors can influence the development of diabetes [12]. Korean diabetic patients are known to have low insulin secretion ability [13], but as the number of overweight and obese diabetic patients increases, insulin resistance is considered a more prominent pathophysiology of diabetes [14]. Severe β-cell dysfunction group had lower odds of chronic kidney disease and severe insulin resistance group had higher odds of carotid artery plaque presence among patients with type 2 diabetes in Korea [15]. However, little research has been conducted on the relationship between IR and β-cell function in relation to multiple diabetic complications among Korean diabetic patients. Therefore, this study aims to investigate the specific associations between IR and β-cell function and multiple diabetic complications, including diabetic nephropathy, diabetic retinopathy, and CVD events, among diabetic patients in Korea, using a common data model (CDM) based on electronic health record (EHR) data.

## Materials and methods

### Data source

We utilized a cohort database sourced from CDM for this study. The CDM database comprises anonymized personal information, such as age, gender, race, and residence, as well as medical records, including physical examinations, diagnoses, laboratory results, treatment modalities, and drug prescriptions. These records were collected during the treatment period at various hospitals and standardized into a common format suitable for multicenter research. Specifically, EHR data from Ewha Womans University Mokdong Hospital, ranging from January 1,

2001, to December 31, 2019, were transformed into the CDM structure based on the Observational Medical Outcomes Partnership CDM version 5.0 using standardized analytical tools [16]. The data were accessed for research purposes from April 14, 2022, to May 5, 2023.

No informed consent was obtained from the study participants, as the data were not initially collected for this research. All patient records from the CDM were anonymized prior to being made accessible for this study. The research was approved by the Institutional Review Board (IRB) of Ewha Womans University Mokdong Hospital.

## Study population

Among the total subjects, there were 9,107 adults aged 20 years or older for whom HOMA-IR and HOMA-β values could be calculated. Our study population comprised 4,113 subjects aged over 20 years who had been diagnosed with diabetes mellitus and had visited Ewha Womans University Mokdong Hospital between January 1, 2001, and December 31, 2019. We sourced these records from the CDM database. Diabetic nephropathy, diabetic retinopathy, and CVD were assessed at both baseline and follow-up examinations.

Diabetes mellitus was identified based on E10-14 codes from the 10th edition of the International Classification of Diseases (ICD-10). We classified subjects as diabetic patients if they had been prescribed oral glucose-lowering medications or insulin, exhibited a glycated hemoglobin (HbA1c) level of 6.5% or higher, or had a fasting plasma glucose level of 126 mg/dL or higher. We excluded subjects with incomplete data (n = 13), those with a history of diabetic nephropathy (n = 139), diabetic retinopathy (n = 104), or CVD (n = 1,615), as well as those who had received insulin prescriptions for more than 90 days from the index date (n = 861). Finally, the study cohort consisted of 2,034 diabetic patients aged over 20 years.

## Outcome variables and covariates

The primary outcomes included time to onset of diabetic nephropathy, diabetic retinopathy, or incidence of CVD. Incidence of CVD was characterized as the first hospital admission with a CVD diagnosis. Due to the limitations of the CDM dataset, which only contains records from a single hospital, it was not feasible to analyze mortality or ascertain cause of death. Cardiovascular events were composite outcomes comprising coronary heart disease, stroke, and peripheral artery disease. Diabetic nephropathy was defined as chronic kidney disease (eGFR < 60 mL/min/1.73 m$^2$ for a duration exceeding 90 days) and/or albuminuria (urinary albumin levels ≥ 300 mg/g creatinine for more than 90 days). Both diabetic retinopathy and CVD were identified using specific ICD-10 codes (E103, E113, or E143 for diabetic retinopathy; I20–I21, I24, I251, or I253–I259 for coronary heart disease; I60–I61 or I63–I64 for stroke; and I739 for peripheral artery disease).

Anthropometric measurements including height and weight were collected for all subjects. BMI was computed using the formula weight (kg)/height (m)$^2$. The HOMA-IR was calculated as the product of fasting insulin level (mIU/L) and fasting glucose level (mmol/L), divided by 22.5. The HOMA-β was calculated using the equation: 20 × fasting insulin (mIU/L)/[fasting glucose (mmol/L)– 3.5] [17].

Drug prescriptions active on the index date were those prescribed for a duration exceeding 30 days. Sociodemographic information, such as age and gender, along with results from physical examinations (BMI and blood pressure), laboratory tests (HbA1c, fasting plasma glucose, fasting plasma insulin, C-peptide, total cholesterol, triglycerides, high-density lipoprotein cholesterol, low-density lipoprotein cholesterol), and treatment modalities (utilization of antidiabetic, antihypertensive, and lipid-lowering therapies) were recorded at the index date.

## Statistical analyses

Baseline characteristics are reported as mean ± standard deviation for continuous variables and as frequency and proportion for categorical variables. Because HOMA-IR and HOMA-β values were not normally distributed, we divided participants into quartiles of HOMA-IR and HOMA-β values. The lowest quartile for HOMA-IR and the highest quartile for HOMA-β served as reference categories. Cox proportional hazards models were employed to evaluate the risk of diabetic nephropathy, diabetic retinopathy, or CVD events. Adjusted hazard ratios (HRs) and 95% confidence intervals were subsequently calculated. We calculated HR in three ways: model 1 was unadjusted, model 2 was adjusted for age and gender, and model 3 was adjusted for age, gender, BMI, and medication use, including antidiabetic, antihypertensive, and lipid-lowering therapies. A sensitivity analysis was also conducted, excluding participants with C-peptide levels below 0.6 mIU/L. We also performed a sensitivity analysis for patients with a follow-up period of less than 2 years.

*P*-values less than 0.05 were deemed statistically significant. All analyses were carried out using SAS software (version 9.4, SAS Institute, Cary, NC).

## Results

Baseline characteristics stratified by quartiles of HOMA-IR and HOMA-β are presented in S1 and S2 Tables, respectively. Fifty percent (n = 1,008) of the total subjects were male and the mean age was 51.6 years. Participants with elevated HOMA-IR tended to exhibit higher values of BMI, total cholesterol, triglycerides, low-density lipoprotein cholesterol, HbA1c, fasting plasma glucose, fasting plasma insulin, C-peptide, and HOMA-β. Conversely, those with lower HOMA-β tended to have higher levels of HbA1c and fasting plasma glucose but lower levels of BMI, fasting plasma insulin, C-peptide, and HOMA-IR.

During the observation period, CVD events were observed in 237 patients with a mean follow-up of two years. The incidence rate of CVD was 10.0% (n = 51) among individuals in the lowest quartile of HOMA-IR, compared to 14.0% (n = 71) in the highest quartile of HOMA-IR. Cerebrovascular events were documented in 146 patients during the study. Specifically, the incidence rate of stroke was 5.3% (n = 27) for those in the lowest HOMA-IR quartile and 9.3% (n = 47) for those in the highest HOMA-IR quartile. Additionally, diabetic nephropathy was reported in 291 patients, with an incidence rate of 13.6% (n = 69) in the lowest quartile of HOMA-IR and 14.8% (n = 75) in the highest quartile (Table 1). Lastly, diabetic retinopathy was identified in 106 patients, with incidence rates of 1.6% (n = 8) for the highest quartile of HOMA-β and 9.8% (n = 50) for the lowest quartile (Table 2).

In diabetic patients belonging to the highest quartile of HOMA-IR, the adjusted hazard ratio for diabetic retinopathy was 0.89 (95% confidence interval [CI], 0.46–1.72; *P* = 0.721) when compared to individuals in the lowest quartile of HOMA-IR (Table 3). Conversely, among diabetic patients in the lowest quartile of HOMA-β, the adjusted hazard ratio was 3.91 (95% CI, 1.80–8.49; *P* = 0.001) for diabetic retinopathy when compared to individuals in the highest quartile of HOMA-β (Table 4).

For diabetic patients in the highest quartile of HOMA-IR, the adjusted hazard ratio for diabetic nephropathy was 1.31 (95% CI, 0.92–1.85; *P* = 0.132) when compared to individuals in the lowest quartile of HOMA-IR (Table 3). Similarly, among diabetic patients in the lowest quartile of HOMA-β, the adjusted hazard ratio for diabetic nephropathy was 1.14 (95% CI, 0.78–1.66; *P* = 0.496) when compared to individuals in the highest quartile of HOMA-β (Table 4).

In diabetic patients in the highest quartile of HOMA-IR, the adjusted hazard ratio for total CVD events was 1.76 (95% CI, 1.20–2.57; *P* = 0.004) when compared to individuals in the

**Table 1. Multiple diabetic complications of participants according to HOMA-IR quartiles.**

| | HOMA-IR quartiles | | | | P for trend |
|---|---|---|---|---|---|
| | 1 (n = 508) | 2 (n = 509) | 3 (n = 509) | 4 (n = 508) | |
| Nephropathy, n (%) | 69 (13.6) | 64 (12.6) | 83 (16.3) | 75 (14.8) | 0.360 |
| Retinopathy, n (%) | 27 (5.3) | 34 (6.7) | 30 (5.9) | 15 (3.0) | 0.047 |
| CVD events, n (%) | 51 (10.0) | 48 (9.4) | 67 (13.2) | 71 (14.0) | 0.057 |
| Coronary events, n (%) | 28 (5.5) | 21 (4.1) | 31 (6.1) | 31 (6.1) | 0.465 |
| Cerebrovascular events, n (%) | 27 (5.3) | 32 (6.3) | 40 (7.9) | 47 (9.3) | 0.076 |
| Peripheral artery diseases, n (%) | 0 (0.0) | 8 (1.6) | 4 (0.8) | 2 (0.4) | 0.018 |

Data are presented as frequencies and percentages.

HOMA-IR, homeostasis model assessment of insulin resistance; CVD, cardiovascular disease.

lowest quartile of HOMA-IR (Table 3). Conversely, among diabetic patients in the lowest quartile of HOMA-β, the adjusted hazard ratio for total CVD events was 0.83 (95% CI, 0.56–1.23; $P = 0.343$) when compared to subjects in the highest quartile of HOMA-β (Table 4).

Additionally, for diabetic patients in the highest quartile of HOMA-IR, the adjusted hazard ratios for stroke and coronary heart disease were 1.92 (95% CI, 1.17–3.16; $P = 0.010$) and 1.35 (95% CI, 0.79–2.32; $P = 0.273$), respectively, compared to those in the lowest quartile of HOMA-IR (Table 3).

In a sensitivity analysis excluding subjects (n = 586) with C-peptide levels below 0.6 mIU/L, results remained consistent (refer to S3 and S4 Tables). Among diabetic patients in the highest quartile of HOMA-IR, the adjusted hazard ratio for total CVD events was 1.80 (95% CI, 1.19–2.72; $P = 0.005$), and for stroke, it was 1.99 (95% CI, 1.19–3.31; $P = 0.009$) when compared with those in the lowest quartile of HOMA-IR. For diabetic patients in the lowest quartile of HOMA-β, the adjusted hazard ratio for diabetic retinopathy was 3.29 (95% CI, 1.46–7.41; $P = 0.004$) relative to those in the highest quartile of HOMA-β. In contrast to the overall diabetic population, diabetic patients in the lowest quartile of HOMA-β exhibited an adjusted hazard ratio of 0.60 for total CVD events (95% CI, 0.39–0.93; $P = 0.021$) compared to those in the highest quartile, after the exclusion of subjects with C-peptide levels below 0.6 mIU/L.

In additional sensitivity analysis with the follow-up period is less than 2 years, results were similar (refer to S5 and S6 Tables). Among diabetic patients in the highest quartile of HOMA-IR, the adjusted hazard ratio for total CVD events was 2.08 (95% CI, 1.30–3.34; $P = 0.002$), and for stroke, it was 2.01 (95% CI, 1.12–3.61; $P = 0.019$) when compared with those in the lowest quartile of HOMA-IR. For diabetic patients in the lowest quartile of

**Table 2. Multiple diabetic complications of participants according to HOMA-β quartiles.**

| | HOMA-β quartiles | | | | P for trend |
|---|---|---|---|---|---|
| | 1 (n = 508) | 2 (n = 509) | 3 (n = 509) | 4 (n = 508) | |
| Nephropathy, n (%) | 75 (14.8) | 81 (15.9) | 83 (16.3) | 52 (10.2) | 0.021 |
| Retinopathy, n (%) | 50 (9.8) | 25 (4.9) | 23 (4.5) | 8 (1.6) | <0.001 |
| CVD events, n (%) | 62 (12.2) | 69 (13.6) | 56 (11.0) | 50 (9.8) | 0.288 |
| Coronary events, n (%) | 32 (6.3) | 26 (5.1) | 27 (5.3) | 26 (5.1) | 0.812 |
| Cerebrovascular events, n (%) | 34 (6.7) | 48 (9.4) | 35 (6.9) | 29 (5.7) | 0.123 |
| Peripheral artery diseases, n (%) | 3 (0.6) | 3 (0.6) | 6 (1.2) | 2 (0.4) | 0.461 |

Data are presented as frequencies and percentages.

HOMA-β, homeostasis model assessment of beta cell function; CVD, cardiovascular disease.

**Table 3. Hazard ratios for diabetic nephropathy, diabetic retinopathy, or cardiovascular events according to HOMA-IR quartiles.**

|  | HOMA-IR quartiles | Unadjusted | | | Adjusted with age and gender | | | Adjusted with all variables | | |
|---|---|---|---|---|---|---|---|---|---|---|
|  |  | HR | 95% CI | P-value | HR | 95% CI | P-value | HR | 95% CI | P-value |
| Diabetic nephropathy | 1 |  |  |  |  |  |  |  |  |  |
|  | 2 | 0.93 | 0.66–1.31 | 0.675 | 1.08 | 0.77–1.52 | 0.654 | 1.11 | 0.78–1.57 | 0.558 |
|  | 3 | 1.25 | 0.91–1.72 | 0.177 | 1.22 | 0.89–1.69 | 0.216 | 1.23 | 0.87–1.72 | 0.239 |
|  | 4 | 1.15 | 0.83–1.59 | 0.409 | 1.20 | 0.87–1.67 | 0.275 | 1.31 | 0.92–1.85 | 0.132 |
| Diabetic retinopathy | 1 |  |  |  |  |  |  |  |  |  |
|  | 2 | 1.33 | 0.80–2.20 | 0.275 | 1.33 | 0.80–2.21 | 0.269 | 1.34 | 0.80–2.24 | 0.266 |
|  | 3 | 1.27 | 0.75–2.14 | 0.369 | 1.24 | 0.74–2.09 | 0.412 | 1.29 | 0.75–2.21 | 0.363 |
|  | 4 | 0.72 | 0.39–1.36 | 0.317 | 0.74 | 0.39–1.38 | 0.340 | 0.89 | 0.46–1.72 | 0.721 |
| Cardiovascular disease | 1 |  |  |  |  |  |  |  |  |  |
|  | 2 | 0.96 | 0.65–1.43 | 0.854 | 1.13 | 0.76–1.68 | 0.543 | 1.22 | 0.82–1.84 | 0.331 |
|  | 3 | 1.44 | 1.00–2.08 | 0.049 | 1.47 | 1.02–2.11 | 0.040 | 1.67 | 1.15–2.44 | 0.007 |
|  | 4 | 1.62 | 1.13–2.33 | 0.009 | 1.74 | 1.21–2.49 | 0.003 | 1.76 | 1.20–2.57 | 0.004 |
| Coronary events | 1 |  |  |  |  |  |  |  |  |  |
|  | 2 | 0.77 | 0.44–1.36 | 0.374 | 0.86 | 0.49–1.51 | 0.591 | 0.88 | 0.49–1.58 | 0.675 |
|  | 3 | 1.25 | 0.75–2.09 | 0.391 | 1.25 | 0.75–2.08 | 0.395 | 1.34 | 0.79–2.28 | 0.277 |
|  | 4 | 1.33 | 0.80–2.23 | 0.273 | 1.40 | 0.84–2.33 | 0.200 | 1.35 | 0.79–2.32 | 0.273 |
| Cerebrovascular events | 1 |  |  |  |  |  |  |  |  |  |
|  | 2 | 1.21 | 0.72–2.01 | 0.472 | 1.49 | 0.89–2.49 | 0.131 | 1.60 | 0.95–2.71 | 0.080 |
|  | 3 | 1.57 | 0.96–2.56 | 0.071 | 1.60 | 0.98–2.61 | 0.059 | 1.77 | 1.07–2.94 | 0.026 |
|  | 4 | 1.93 | 1.20–3.11 | 0.006 | 2.02 | 1.26–3.25 | 0.004 | 1.92 | 1.17–3.16 | 0.010 |

Hazard ratios were adjusted for age, gender, body mass index, and prescriptions for antidiabetic, antihypertensive, and lipid-lowering therapies.

HOMA-IR, homeostasis model assessment of insulin resistance; HR, hazard ratio; CI, confidence interval.

HOMA-β, there was no statistically significant difference, but the adjusted hazard ratio for diabetic retinopathy was 6.28 (95% CI, 0.77–51.27; *P* = 0.086) relative to those in the highest quartile of HOMA-β.

## Discussion

In our study of Korean diabetic patients using the CDM program, we found that IR and β-cell function exhibited distinct relationships with diabetic complications. Specifically, higher HOMA-IR values were associated with elevated CVD risk, whereas lower HOMA-β values correlated with a greater risk of retinopathy.

Several prior studies have similarly evaluated the clinical features and complications risks in diabetic populations using novel diabetes classifications across diverse ethnic groups. For instance, a U.S. study involving 712 diabetic participants classified into five different diabetes mellitus subgroups found that the MARD subgroup had greater CVD-related mortality than the MOD subgroup. Furthermore, both SAID and SIDD subgroups demonstrated a heightened risk of diabetic retinopathy compared to the MARD subgroup [18]. IR, quantified using HOMA-IR, emerged as an independent predictor of both existing and newly developed CVD, including coronary, cerebrovascular, and peripheral vascular diseases in 1,326 type 2 diabetic subjects participating in the Verona Diabetes Complications Study [6]. Among 93,690 Chinese adults who were not on antidiabetic medications, a higher HOMA-IR was independently associated with a higher prevalence of established cardiometabolic disorders [10]. Similarly, a reduced β-cell function, as measured by the insulinogenic index, was associated with

**Table 4. Hazard ratios for diabetic nephropathy, diabetic retinopathy, or cardiovascular events according to HOMA-β quartiles.**

| | HOMA-β quartiles | Unadjusted | | | Adjusted with age and gender | | | Adjusted with all variables | | |
|---|---|---|---|---|---|---|---|---|---|---|
| | | HR | 95% CI | P-value | HR | 95% CI | P-value | HR | 95% CI | P-value |
| Diabetic nephropathy | 1 | 1.40 | 0.98–2.00 | 0.062 | 1.12 | 0.78–1.59 | 0.543 | 1.14 | 0.78–1.66 | 0.496 |
| | 2 | 1.54 | 1.09–2.18 | 0.015 | 1.19 | 0.84–1.69 | 0.318 | 1.20 | 0.84–1.71 | 0.313 |
| | 3 | 1.60 | 1.13–2.26 | 0.008 | 1.25 | 0.89–1.77 | 0.201 | 1.17 | 0.83–1.66 | 0.379 |
| | 4 | | | | | | | | | |
| Diabetic retinopathy | 1 | 4.84 | 2.29–10.22 | < .001 | 4.21 | 1.98–8.94 | < .001 | 3.91 | 1.80–8.49 | 0.001 |
| | 2 | 2.65 | 1.20–5.89 | 0.016 | 2.29 | 1.03–5.12 | 0.043 | 2.06 | 0.92–4.64 | 0.080 |
| | 3 | 2.48 | 1.11–5.54 | 0.027 | 2.29 | 1.02–5.15 | 0.044 | 2.02 | 0.90–4.55 | 0.089 |
| | 4 | | | | | | | | | |
| Cardiovascular disease | 1 | 1.14 | 0.78–1.65 | 0.501 | 0.91 | 0.63–1.33 | 0.626 | 0.83 | 0.56–1.23 | 0.343 |
| | 2 | 1.32 | 0.92–1.90 | 0.137 | 1.02 | 0.71–1.48 | 0.903 | 1.03 | 0.71–1.50 | 0.862 |
| | 3 | 1.07 | 0.73–1.57 | 0.728 | 0.89 | 0.61–1.31 | 0.560 | 0.94 | 0.64–1.38 | 0.762 |
| | 4 | | | | | | | | | |
| Coronary events | 1 | 1.10 | 0.65–1.84 | 0.730 | 0.85 | 0.50–1.43 | 0.529 | 0.81 | 0.47–1.41 | 0.459 |
| | 2 | 0.92 | 0.53–1.59 | 0.762 | 0.69 | 0.40–1.20 | 0.189 | 0.70 | 0.40–1.23 | 0.217 |
| | 3 | 0.98 | 0.57–1.69 | 0.953 | 0.82 | 0.48–1.41 | 0.471 | 0.85 | 0.49–1.46 | 0.545 |
| | 4 | | | | | | | | | |
| Cerebrovascular events | 1 | 1.08 | 0.66–1.78 | 0.759 | 0.90 | 0.54–1.48 | 0.665 | 0.83 | 0.49–1.40 | 0.474 |
| | 2 | 1.59 | 1.00–2.53 | 0.048 | 1.28 | 0.80–2.03 | 0.301 | 1.31 | 0.81–2.10 | 0.270 |
| | 3 | 1.16 | 0.71–1.90 | 0.549 | 0.97 | 0.59–1.58 | 0.895 | 0.98 | 0.60–1.62 | 0.947 |
| | 4 | | | | | | | | | |

Hazard ratios were adjusted for age, gender, body mass index, and prescriptions for antidiabetic, antihypertensive, and lipid-lowering therapies.

HOMA-β, homeostasis model assessment of beta cell function; HR, hazard ratio; CI, confidence interval.

albuminuria, while impaired insulin sensitivity, gauged through the Matsuda insulin sensitivity index, was associated with elevated coronary artery calcification in 672 type 2 diabetic patients without cardiovascular or renal disease in the Penn Diabetes Heart Study [19]. In 544 newly diagnosed Caucasian subjects with type 2 diabetes who underwent a standardized meal tolerance test, diabetic retinopathy was tied to reduced β-cell responsiveness resulting from β-cell failure [7]. Our findings align with these previous studies, indicating that in a Korean diabetic population, lower β-cell function increases the risk of diabetic retinopathy, while higher IR amplifies CVD risk. Importantly, our sensitivity analysis, which excluded subjects with C-peptide levels below 0.6 mIU/L, suggested that the observed decrease in CVD risk among diabetic patients with lower HOMA-β may be associated with low IR levels in the context of compromised β-cell function.

In our study, we were unable to establish a link between IR or β-cell function and the risk of diabetic kidney disease. This outcome contrasts with previous studies that utilized the novel diabetes classification system. For instance, among 8,980 newly diagnosed diabetic patients in Sweden, the SIRD subgroup demonstrated a heightened risk for diabetic kidney disease, while the SIDD subgroup showed an increased risk for retinopathy [5]. Similarly, in a retrospective cohort study conducted in Japan, which included 1,255 diabetic patients, those in the SAID or SIDD subgroups were found to be at a higher risk for diabetic retinopathy, whereas patients in the SIRD subgroup were more susceptible to diabetic kidney disease [11]. One plausible explanation for these inconsistencies could lie in the differing definitions of diabetic kidney disease across the studies, particularly in the criteria for albuminuria. In our research, albuminuria was identified as a urinary albumin to creatinine ratio equal to or exceeding 300 mg/g

creatinine for a duration exceeding 90 days. In contrast, the Swedish study classified albuminuria as an albumin excretion rate of 200 $\mu$g/min or higher, 300 mg/day or higher, or an albumin to creatinine ratio of 25 mg/mmol or higher for men and 35 mg/mmol or higher for women on at least two of three consecutive visits [5]. Meanwhile, the Japanese study defined diabetic kidney disease as either chronic kidney disease and/or proteinuria, with proteinuria identified as a 1+ result on dipstick urine tests maintained for over 90 days [11]. Genetic and environmental factors may affect the development of diabetes [12], and differences in genetic and environmental factors by race may have affected different results across studies.

Traditionally, β-cell dysfunction was thought to be the primary factor in the development of diabetes, and Asian populations, including Koreans, were observed to develop diabetes with lower rates of obesity compared to Caucasians. For instance, in Korean subjects who underwent an oral glucose tolerance test, impairment in early-phase insulin secretion, assessed via the insulinogenic index, was suggested as the initial abnormality leading to type 2 diabetes [20]. However, a cross-sectional study conducted between 2009 and 2010, which included 1,314 Korean patients older than 18 years with newly diagnosed diabetes, revealed that 59.5% of the subjects exhibited IR, and 20.2% had moderate to severe defects in insulin secretion as assessed by C-peptide levels [21]. With the rising prevalence of obesity in Korea, largely attributed to shifts toward a high-fat diet and reduced physical activity, there has been a corresponding increase in the incidence of type 2 diabetes, particularly among younger individuals. Analysis of nationwide cross-sectional data revealed that individuals diagnosed with diabetes before the age of 40 had significantly higher levels of HOMA-IR and reduced β-cell function compared to those diagnosed after the age of 65, among 912 participants with newly diagnosed type 2 diabetes in Korea [22]. Given the changing landscape of type 2 diabetes pathogenesis in Korea, shifting from insulin deficiency to IR, it becomes crucial to assess both IR and β-cell function during the treatment of diabetic patients in order to intervene early and effectively manage diabetic complications.

This represents the first study to examine the distinct relationships between IR and β-cell function with multiple complications of diabetes, including CVD events, in a Korean diabetic population. Our data set, derived from CDM records, spans a substantial recruitment period from January 1, 2001, to December 31, 2019, and includes both male and female participants from various age groups over 20 years old. Thus, the findings of this study are generalizable to the Korean diabetic population. Additionally, we adjusted for multiple confounding variables, such as age, gender, BMI, and prescriptions for diabetes, hypertension, and dyslipidemia treatment, to mitigate their influence on the outcomes.

However, our study is not without limitations. We did not employ gold standard methods, such as the hyperinsulinemic-euglycemic clamp test, for assessing β-cell function. Furthermore, the study lacks evaluations of the relationships between IR and β-cell function with treatment response and mortality rates among diabetic patients. A previous study of German patients newly diagnosed with diabetes found that the prevalence of nonalcoholic fatty liver disease and diabetic neuropathy varied across specific diabetes populations [23]. Ectopic fat distribution is known to be a significant driver of IR and is believed to impact insulin secretion [1]. However, we did not explore the association of IR and β-cell function with other diabetic complications like non-alcoholic fatty liver disease or diabetic polyneuropathy. Since subjects for whom HOMA-IR and HOMA-β values could not be measured were excluded from the study, selection bias may have affected the study results. Additional research is needed to investigate the relationships between IR, β-cell function, and other diabetic complications, as well as treatment response.

Lastly, the retrospective nature of our study restricts our ability to elucidate the mechanisms underlying the differential associations between IR and β-cell function with the array of

complications observed in diabetic patients. Because IR was known to be associated with several risk factors for CVD including hyperglycemia, dyslipidemia, and hypertension [9, 10], so in this study, patients with diabetes and increased IR may have an increased risk of CVD. Previous epidemiologic studies have shown that β-cell dysfunction was associated with an increased risk of severe diabetic retinopathy [24–26], but the mechanism is unclear. β-cell dysfunction may be a surrogate marker of diabetes duration and more progressive forms of diabetes.

In conclusion, diminished β-cell function, as estimated by HOMA-β, was associated with an elevated risk of developing diabetic retinopathy. Conversely, elevated IR, as quantified by HOMA-IR, was associated with an increased likelihood of CVD events among Korean individuals with diabetes. In order to prevent and manage complications in Korean diabetic patients, it is necessary to understand the pathophysiology of Korean diabetic patients. It is also necessary to identify subgroups vulnerable to diabetic complications and thoroughly screen them in advance to manage complications. Assessments of IR and β-cell function can serve as valuable predictors for complications in this patient population. Tailored therapeutic strategies may be warranted based on variations in IR and β-cell function. Specifically, individuals with pronounced IR should prioritize preventive measures against CVD, while those with significant β-cell dysfunction may gain from early, intensive surveillance for diabetic retinopathy.

## Supporting information

**S1 Table. Baseline clinical characteristics of participants according to HOMA-IR quartiles.** Data are presented as mean ± standard deviation or as frequency and proportion. HOMA-IR, homeostasis model assessment of insulin resistance; BMI, body mass index; TC, total cholesterol; TG, triglycerides; HDL-C, high-density lipoprotein cholesterol; LDL-C, low-density lipoprotein cholesterol; HbA1c, glycated hemoglobin; HOMA-β, homeostasis model assessment of beta cell function; SBP, systolic blood pressure; DBP, diastolic blood pressure.
(DOCX)

**S2 Table. Baseline clinical characteristics of participants according to HOMA-β quartiles.** Data are presented as mean ± standard deviation or as frequency and proportion. HOMA-β, homeostasis model assessment of beta cell function; BMI, body mass index; TC, total cholesterol; TG, triglycerides; HDL-C, high-density lipoprotein cholesterol; LDL-C, low-density lipoprotein cholesterol; HbA1c, glycated hemoglobin; HOMA-IR, homeostasis model assessment of insulin resistance; SBP, systolic blood pressure; DBP, diastolic blood pressure.
(DOCX)

**S3 Table. Hazard ratios for diabetic nephropathy, diabetic retinopathy, or cardiovascular events according to HOMA-IR quartiles, excluding subjects with C-peptide below 0.6 mIU/L.** Hazard ratios were adjusted for age, gender, body mass index, and prescriptions for antidiabetic, antihypertensive, and lipid-lowering therapies. HOMA-IR, homeostasis model assessment of insulin resistance; HR, hazard ratio; CI, confidence interval.
(DOCX)

**S4 Table. Hazard ratios for diabetic nephropathy, diabetic retinopathy, or cardiovascular events according to HOMA-β quartiles, excluding subjects with C-peptide below 0.6 mIU/L.** Hazard ratios were adjusted for age, gender, body mass index, and prescriptions for antidiabetic, antihypertensive, and lipid-lowering therapies. HOMA-β, homeostasis model assessment of beta cell function; HR, hazard ratio; CI, confidence interval.
(DOCX)

**S5 Table. Hazard ratios for diabetic nephropathy, diabetic retinopathy, or cardiovascular events according to HOMA-IR quartiles with the follow-up period is less than 2 years.** Hazard ratios were adjusted for age, gender, body mass index, and prescriptions for antidiabetic, antihypertensive, and lipid-lowering therapies. HOMA-IR, homeostasis model assessment of insulin resistance; HR, hazard ratio; CI, confidence interval.
(DOCX)

**S6 Table. Hazard ratios for diabetic nephropathy, diabetic retinopathy, or cardiovascular events according to HOMA-β quartiles with the follow-up period is less than 2 years.** Hazard ratios were adjusted for age, gender, body mass index, and prescriptions for antidiabetic, antihypertensive, and lipid-lowering therapies. HOMA-β, homeostasis model assessment of beta cell function; HR, hazard ratio; CI, confidence interval.
(DOCX)

## Author Contributions

**Conceptualization:** Do Kyeong Song, Hyejin Lee.

**Funding acquisition:** Do Kyeong Song.

**Investigation:** Do Kyeong Song, Young Sun Hong, Yeon-Ah Sung.

**Methodology:** Do Kyeong Song, Young Sun Hong, Yeon-Ah Sung, Hyejin Lee.

**Project administration:** Hyejin Lee.

**Supervision:** Young Sun Hong, Yeon-Ah Sung, Hyejin Lee.

**Writing – original draft:** Do Kyeong Song.

**Writing – review & editing:** Hyejin Lee.

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
