## [Decision Letter · Decision Letter 0]

16 Aug 2024

PONE-D-24-25485Effects of insulin resistance and β-cell function on diabetic complications in Korean diabetic patientsPLOS ONE

Dear Dr. Lee,

Thank you for submitting your manuscript to PLOS ONE. After careful consideration, we feel that it has merit but does not fully meet PLOS ONE’s publication criteria as it currently stands. Therefore, we invite you to submit a revised version of the manuscript that addresses the points raised during the review process.

We look forward to receiving your revised manuscript.

Kind regards,

Mohammad Reza Mahmoodi, Ph.D.

Academic Editor

PLOS ONE

Journal Requirements:

"No potential conflict of interest relevant to this article was reported."

Reviewers' comments:

Reviewer's Responses to Questions

**Comments to the Author**

1. Is the manuscript technically sound, and do the data support the conclusions?

Reviewer #1: Yes

Reviewer #2: Yes

Reviewer #3: Partly

2. Has the statistical analysis been performed appropriately and rigorously? 

Reviewer #1: Yes

Reviewer #2: No

Reviewer #3: Yes

3. Have the authors made all data underlying the findings in their manuscript fully available?

Reviewer #1: Yes

Reviewer #2: Yes

Reviewer #3: No

4. Is the manuscript presented in an intelligible fashion and written in standard English?

Reviewer #1: Yes

Reviewer #2: Yes

Reviewer #3: Yes

5. Review Comments to the Author

Reviewer #1: In the manuscript titled "Effects of insulin resistance and β-cell function on diabetic complications in Korean diabetic patients," the authors explore how insulin resistance and β-cell function relate to diabetic complications in Korean patients. This research question is compelling, and the findings are well-supported, offering potential clinical significance. With some adjustments for clarity and detail, this work should be ready for publication.

•

The abstract could be improved by including more specifics about the clinical implications of the findings.

•

The introduction does a good job of outlining the complex nature of diabetes. However, it could be clearer about the specific gaps in the current research that this study addresses, especially regarding Korean diabetic patients. It would help to emphasize why it's important to study this particular group.

•

The methods section is detailed, explaining the use of a common data model (CDM) and defining how insulin resistance and β-cell function were measured. It would be useful to include more details about how patients were selected and the specific statistical methods used to calculate hazard ratios. Addressing potential biases and how they were mitigated would strengthen this section. It’s also important to explain how patient data was anonymized and how any missing data was managed.

• For clarity, some sentences could be simplified. For instance, instead of "While approximately 75–85% of diabetic individuals fit the traditional criteria for type 2 diabetes...," you could say, "Approximately 75–85% of diabetic patients meet the traditional criteria for type 2 diabetes, but many show characteristics of both insulin resistance and β-cell dysfunction."

•

The discussion should compare the findings with previous studies, particularly those involving other ethnic groups, to highlight the unique aspects of the results. The conclusion could more clearly outline the practical implications, such as how these findings might influence screening or treatment decisions based on insulin resistance and β-cell function. Discussing potential reasons behind the associations found, such as cultural or genetic factors specific to Korean populations, would add depth.

Reviewer #2: Authors looked into the dose-response relationship of beta-cell function and diabetes complications in diabetic individuals.

Abstract is poor. Patient characteristics must be stated in the results section not method. A brief explanation of your data source will be useful in the methods. Also confidence intervals for adjusted hazard ratios must be reported.

Second paragraph of the introduction was too long. I think you can summarize it and talk less about the swedish diabetes classification and more about the importance of beta-cell function and diabetes complications.

Paragraph 3 in introduction: Aim of study is not clear. Why, despite the other similar papers in different countries, it is important to investigate the relationship in Korean population?

Your follow-up duration is vague. You did not report it anywhere as I check. Please calculate it. Moreover, I also suggest subgroup analysis on those with a follow-up duration <2 years (Short-term outcome).

Did you include all the variables in your regression model for adjustment? Or just the statistically significant variables. I suggest to perform three regression models to have a better comparison: model 1= unadjusted, model 2= age and sex adjusted, model 3= adjusted with all confounding variables.

Baseline characteristics of the patients must be stated using numbers (at least for important variables such as age, sex,

Results were poorly organized. I think separation into sections based on different diabetic complications such as retinopathy, nephropathy, and CVD, and then report each relevant finding in its specific section might be suitable.

Again all the confidence intervals are missed. Please report. Also provide p-values and statistical tests for tables 1&2.

Discussion is well-written. However, adding a paragraph about the possible molecular mechanisms underneath the association of diabetic retinopathy or CVD risk with beta-cell function and insulin resistance.

Reviewer #3: Review Comment:

1- The study investigates the associations between insulin resistance (IR), β-cell function, and diabetic complications using a cohort from a common data model (CDM) database. While the topic is relevant, the study's novelty is not well articulated in the abstract, which raises concerns about its contribution to the existing body of knowledge.

2- Methodology: The study employs appropriate methodologies, including the calculation of HOMA-IR and HOMA-β, and robust statistical analyses using Cox proportional hazards models. The inclusion of anthropometric measurements, laboratory tests, and detailed statistical adjustments for variables such as age, gender, BMI, and medication use strengthens the study’s methodology. However, the lack of demographic details such as age, gender, and race are significant omissions that hinder the assessment of the study’s generalizability. The exclusion criteria, particularly the exclusion of subjects with incomplete data or prior complications, need more justification to ensure that the study's findings are not biased. Additionally, the stratification of patients based on quartiles of HOMA-IR and HOMA-β is appropriate, but the rationale behind these specific stratifications and reference categories should be clarified.

3- Ethical Considerations: The authors mention that the study received approval from the hosting institute and used anonymized data, which addresses privacy concerns. However, the absence of informed consent is a critical issue. The abstract does not clearly state whether this research qualifies as a split study, where data originally collected for a different purpose are repurposed. It’s important that the authors clarify this point and detail how ethical approval covered the secondary use of such data.

4- References: The study cites only 15 references, which seems relatively few given the complexity of the topic. This limited reference list may indicate that the study has not fully engaged with the breadth of existing literature, particularly in a field as well-studied as diabetes and its complications. A more comprehensive review of relevant studies would strengthen the contextualization and credibility of the findings.

5- Conclusion: While the study potentially offers valuable insights, several aspects need further clarification and expansion. The abstract should clearly articulate the novelty of the research, include demographic information, address ethical concerns related to the possible use of previously collected data, and consider expanding the reference list to demonstrate a thorough engagement with the existing literature. These major revisions would significantly enhance the rigor and transparency of the study.

6. PLOS authors have the option to publish the peer review history of their article (what does this mean?). If published, this will include your full peer review and any attached files.

Reviewer #1: **Yes: **Solaleh Emamgholipour

Reviewer #2: No

Reviewer #3: No

---

## [Author Response · Author response to Decision Letter 0]

20 Sep 2024

Journal Requirements:

→ Thanks for your comment. We revised the manuscript to conform to PLOS ONE's style requirements.

"No potential conflict of interest relevant to this article was reported."

→ Thank you for your consideration. "The authors have declared that no competing interests exist."

→ Thanks for your comment. Our submission contains all raw data required to replicate the results of our study.

Reviewers' comments:

Reviewer's Responses to Questions

Comments to the Author

1. Is the manuscript technically sound, and do the data support the conclusions?

Reviewer #1: Yes

Reviewer #2: Yes

Reviewer #3: Partly

2. Has the statistical analysis been performed appropriately and rigorously?

Reviewer #1: Yes

Reviewer #2: No

Reviewer #3: Yes

3. Have the authors made all data underlying the findings in their manuscript fully available?

Reviewer #1: Yes

Reviewer #2: Yes

Reviewer #3: No

4. Is the manuscript presented in an intelligible fashion and written in standard English?

Reviewer #1: Yes

Reviewer #2: Yes

Reviewer #3: Yes

5. Review Comments to the Author

Reviewer #1: In the manuscript titled "Effects of insulin resistance and β-cell function on diabetic complications in Korean diabetic patients," the authors explore how insulin resistance and β-cell function relate to diabetic complications in Korean patients. This research question is compelling, and the findings are well-supported, offering potential clinical significance. With some adjustments for clarity and detail, this work should be ready for publication.

•

The abstract could be improved by including more specifics about the clinical implications of the findings.

→ Thanks for your comment. “Individuals with pronounced IR should prioritize CVD prevention measures, and those with significant β-cell dysfunction may benefit from early, intensive surveillance for diabetic retinopathy.” was inserted in the Abstract (line 40-42).

The introduction does a good job of outlining the complex nature of diabetes. However, it could be clearer about the specific gaps in the current research that this study addresses, especially regarding Korean diabetic patients. It would help to emphasize why it's important to study this particular group.

→ Thanks for your comment. “Severe β-cell dysfunction group had lower odds of chronic kidney disease and severe insulin resistance group had higher odds of carotid artery plaque presence among patients with type 2 diabetes in Korea. However, little research has been conducted on the relationship between IR and β-cell function in relation to multiple diabetic complications among Korean diabetic patients.” were inserted in the Introduction section (line 75-79).

The methods section is detailed, explaining the use of a common data model (CDM) and defining how insulin resistance and β-cell function were measured. It would be useful to include more details about how patients were selected and the specific statistical methods used to calculate hazard ratios. Addressing potential biases and how they were mitigated would strengthen this section. It’s also important to explain how patient data was anonymized and how any missing data was managed.

→ “Among the total subjects, there were 9,107 adults aged 20 years or older for whom HOMA-IR and HOMA-β values could be calculated. Our study population comprised 4,113 subjects aged over 20 years who had been diagnosed with diabetes mellitus and had visited Ewha Womans University Mokdong Hospital between January 1, 2001, and December 31, 2019. We sourced these records from the CDM database.” (line 101-105)

“We excluded subjects with incomplete data (n = 13), those with a history of diabetic nephropathy (n = 139), diabetic retinopathy (n = 104), or CVD (n = 1,615), as well as those who had received insulin prescriptions for more than 90 days from the index date (n = 861). Finally, the study cohort consisted of 2,034 diabetic patients aged over 20 years.” (line 114)

“We calculated HR in three ways: model 1 was unadjusted, model 2 was adjusted for age and gender, and model 3 was adjusted for age, gender, BMI, and medication use, including antidiabetic, antihypertensive, and lipid-lowering therapies.” were inserted in the Methods section (line 148-151). 

“Since subjects for whom HOMA-IR and HOMA-β values could not be measured were excluded from the study, selection bias may have affected the study results.” was inserted in the Discussion section (line 298-300).

• For clarity, some sentences could be simplified. For instance, instead of "While approximately 75–85% of diabetic individuals fit the traditional criteria for type 2 diabetes...," you could say, "Approximately 75–85% of diabetic patients meet the traditional criteria for type 2 diabetes, but many show characteristics of both insulin resistance and β-cell dysfunction."

→ Thanks for your comment. As you recommended, we changed the sentences to make it more concise.

The discussion should compare the findings with previous studies, particularly those involving other ethnic groups, to highlight the unique aspects of the results. The conclusion could more clearly outline the practical implications, such as how these findings might influence screening or treatment decisions based on insulin resistance and β-cell function. Discussing potential reasons behind the associations found, such as cultural or genetic factors specific to Korean populations, would add depth.

→ Thanks for your comment. 

“A previous study of German patients newly diagnosed with diabetes found that the prevalence of nonalcoholic fatty liver disease and diabetic neuropathy varied across specific diabetes populations.” (line 293-295)

“Because IR was known to be associated with several risk factors for CVD including hyperglycemia, dyslipidemia, and hypertension [9, 10], so in this study, patients with diabetes and increased IR may have an increased risk of CVD. Previous epidemiologic studies have shown that β-cell dysfunction was associated with an increased risk of severe diabetic retinopathy [24-26], but the mechanism is unclear. β-cell dysfunction may be a surrogate marker of diabetes duration and more progressive forms of diabetes.” (line 305-310)

“In order to prevent and manage complications in Korean diabetic patients, it is necessary to understand the pathophysiology of Korean diabetic patients. It is also necessary to identify subgroups vulnerable to diabetic complications and thoroughly screen them in advance to manage complications.” (line 314-317)

“Genetic and environmental factors may affect the development of diabetes, and differences in genetic and environmental factors by race may have affected different results across studies.” was inserted in the Discussion section (line 261-263).

Reviewer #2: Authors looked into the dose-response relationship of beta-cell function and diabetes complications in diabetic individuals.

Abstract is poor. Patient characteristics must be stated in the results section not method. A brief explanation of your data source will be useful in the methods. Also confidence intervals for adjusted hazard ratios must be reported.

→ Thanks for your comment. “The study cohort consisted of 2,034 diabetic patients aged over 20 years who visited EUMC between January 2001 and December 2019.” was stated in the results section. We included confidence intervals for adjusted hazard ratios in the abstract.

Second paragraph of the introduction was too long. I think you can summarize it and talk less about the swedish diabetes classification and more about the importance of beta-cell function and diabetes complications.

→ Thanks for your comment. We summarize second paragraph of the introduction and talk less about the swedish diabetes classification.

Paragraph 3 in introduction: Aim of study is not clear. Why, despite the other similar papers in different countries, it is important to investigate the relationship in Korean population?

→ Thanks for your comment. “Genetic and environmental factors can influence the development of diabetes. Korean diabetic patients are known to have low insulin secretion ability, but as the number of overweight and obese diabetic patients increases, insulin resistance is considered a more prominent pathophysiology of diabetes.” were inserted in the Introduction section (line 71-74).

Your follow-up duration is vague. You did not report it anywhere as I check. Please calculate it. Moreover, I also suggest subgroup analysis on those with a follow-up duration <2 years (Short-term outcome).

→ Thanks for your comment. “with a mean follow-up of two years.” was inserted in the Results section. “In additional sensitivity analysis with the follow-up period is less than 2 years, results were similar (refer to Supplemental Tables S5 and S6). Among diabetic patients in the highest quartile of HOMA-IR, the adjusted hazard ratio for total CVD events was 2.08 (95% CI, 1.30-3.34; P = 0.002), and for stroke, it was 2.01 (95% CI, 1.12-3.61; P = 0.019) when compared with those in the lowest quartile of HOMA-IR. For diabetic patients in the lowest quartile of HOMA-β, there was no statistically significant difference, but the adjusted hazard ratio for diabetic retinopathy was 6.28 (95% CI, 0.77-51.27; P = 0.086) relative to those in the highest quartile of HOMA-β.” were inserted in the Results section (line 207-214).

Did you include all the variables in your regression model for adjustment? Or just the statistically significant variables. I suggest to perform three regression models to have a better comparison: model 1= unadjusted, model 2= age and sex adjusted, model 3= adjusted with all confounding variables.

→ Thanks for your comment. Three regression models are shown in Tables 3 and 4.

Baseline characteristics of the patients must be stated using numbers (at least for important variables such as age, sex,

→ Thanks for your comment. Baseline characteristics of the patients are shown in Tables 1 and 2. “Fifty percent (n = 1,008) of the total subjects were male and the mean age was 51.6 years.” was inserted in the Results section (line 159-160).

Results were poorly organized. I think separation into sections based on different diabetic complications such as retinopathy, nephropathy, and CVD, and then report each relevant finding in its specific section might be suitable.

→ Thanks for your comment. We divided the sections based on different diabetic complications such as retinopathy, nephropathy, and CVD.

Again all the confidence intervals are missed. Please report. Also provide p-values and statistical tests for tables 1&2.

→ Thanks for your comment. We revised the results by adding confidence intervals. And we provided p-values and statistical tests for Tables 1 and 2.

Discussion is well-written. However, adding a paragraph about the possible molecular mechanisms underneath the association of diabetic retinopathy or CVD risk with beta-cell function and insulin resistance.

→ Thanks for your comment. “Because IR was known to be associated with several risk factors for CVD including hyperglycemia, dyslipidemia, and hypertension [9, 10], so in this study, patients with diabetes and increased IR may have an increased risk of CVD. Previous epidemiologic studies have shown that β-cell dysfunction was associated with an increased risk of severe diabetic retinopathy, but the mechanism is unclear. β-cell dysfunction may be a surrogate marker of diabetes duration and more progressive forms of diabetes.” were inserted in the Discussion section (line 304-310). 

Reviewer #3: Review Comment:

1- The study investigates the associations between insulin resistance (IR), β-cell function, and diabetic complications using a cohort from a common data model (CDM) database. While the topic is relevant, the study's novelty is not well articulated in the abstract, which raises concerns about

---

## [Decision Letter · Decision Letter 1]

8 Oct 2024

Effects of insulin resistance and β-cell function on diabetic complications in Korean diabetic patients

PONE-D-24-25485R1

Dear Dr. Lee,

We’re pleased to inform you that your manuscript has been judged scientifically suitable for publication and will be formally accepted for publication once it meets all outstanding technical requirements.

Kind regards,

Mohammad Reza Mahmoodi, Ph.D.

Academic Editor

PLOS ONE

Additional Editor Comments (optional):

Reviewers' comments:

Reviewer's Responses to Questions

**Comments to the Author**

1. If the authors have adequately addressed your comments raised in a previous round of review and you feel that this manuscript is now acceptable for publication, you may indicate that here to bypass the “Comments to the Author” section, enter your conflict of interest statement in the “Confidential to Editor” section, and submit your "Accept" recommendation.

Reviewer #1: All comments have been addressed

Reviewer #2: All comments have been addressed

Reviewer #3: All comments have been addressed

2. Is the manuscript technically sound, and do the data support the conclusions?

Reviewer #1: Yes

Reviewer #2: Yes

Reviewer #3: Yes

3. Has the statistical analysis been performed appropriately and rigorously? 

Reviewer #1: Yes

Reviewer #2: Yes

Reviewer #3: Yes

4. Have the authors made all data underlying the findings in their manuscript fully available?

Reviewer #1: Yes

Reviewer #2: Yes

Reviewer #3: Yes

5. Is the manuscript presented in an intelligible fashion and written in standard English?

Reviewer #1: Yes

Reviewer #2: Yes

Reviewer #3: Yes

6. Review Comments to the Author

Reviewer #1: The authors have thoroughly addressed the reviewers' comments, improving the study's methods, clarity, and strength. The manuscript is now well-organized with better discussion and interpretation, making it more suitable for publication.

Reviewer #2: Thanks for your precise and reasonable responses. I have no comments to add. I recommend the acceptance.

Reviewer #3: (No Response)

7. PLOS authors have the option to publish the peer review history of their article (what does this mean?). If published, this will include your full peer review and any attached files.

Reviewer #1: **Yes: **Solaleh Emamgholipour

Reviewer #2: No

Reviewer #3: No

---

## [Editor Report · Acceptance letter]

11 Oct 2024

PONE-D-24-25485R1 

PLOS ONE

Dear Dr. Lee, 

I'm pleased to inform you that your manuscript has been deemed suitable for publication in PLOS ONE. Congratulations! Your manuscript is now being handed over to our production team.

Kind regards, 

on behalf of

Dr. Mohammad Reza Mahmoodi 

Academic Editor

PLOS ONE